# Causes and Risk Factors of Breast Cancer, What Do We Know for Sure? An Evidence Synthesis of Systematic Reviews and Meta-Analyses

**DOI:** 10.3390/cancers16081583

**Published:** 2024-04-20

**Authors:** Borghild Løyland, Ida Hellum Sandbekken, Ellen Karine Grov, Inger Utne

**Affiliations:** Department of Nursing and Health Promotion, Faculty of Health Sciences, Oslo Metropolitan University, 0130 Oslo, Norway; idahan@oslomet.no (I.H.S.); ellgro@oslomet.no (E.K.G.); inger@oslomet.no (I.U.)

**Keywords:** breast cancer, women, causes, risk factors, systematic review

## Abstract

**Simple Summary:**

Breast cancer affects women in every country in the world and is the type of cancer that causes the most deaths among women. The disease is, nevertheless, one of the very few where it is women with higher education and a higher socio-economic status who are primarily affected. Most of those affected live in developed countries in Europe and North America. The purpose of this study was to gain more knowledge about what could be said with certainty to be the causes of breast cancer in women. Could it, for example, be something to do with the way women in these countries live their lives? Unfortunately, an extensive literature review did not give us very good answers. We still know little about the causes of breast cancer in women.

**Abstract:**

Breast cancer affected more than 2.3 million women in 2022 and is the most diagnosed cancer among women worldwide. The incidence rates are greater in developed regions and are significantly higher among women with higher education and socioeconomic status. Therefore, it is reasonable to assume that the way women live their lives may impact their risk of being diagnosed with breast cancer. This systematic review aimed to identify what is known about the causes and risk factors of breast cancer, excluding genetic causes. A comprehensive systematic search identified 2387 systematic reviews, 122 were included and six overall themes identified. In our “top list” with the 36 most important findings, a study of breast density had the highest effect size for increasing the risk of breast cancer, and a high sex-hormone-binding globulin level was the most protective factor. Many of the included studies investigating the same topics had conflicting results. The conclusion from this evidence synthesis reveals a lack of consensus of factors associated with the causes and risk of breast cancer. These findings suggest that recommendations about lifestyle and breast cancer should be made with caution.

## 1. Introduction

Breast cancer affected more than 2.3 million women in 2022 and is the most diagnosed cancer among women worldwide [1,2]. Breast cancer is also the number one cause of death from cancer in women; 670,000 women died from this disease in 2022 [2]. That is approximately 7% of all cancer deaths worldwide. The incidence rates are greater than 80 per 100,000 in developed regions of the world, such as Europe and North America, Australia, and New Zealand, and less than 40 per 100,000 in developing countries [3,4]. However, statistics from some developing countries may be attributed to underreporting [5]. The differences in incidence between countries could be due to changes in exposure to environmental risk factors, behavior, and lifestyle factors of different population groups [6]. These facts provide good reasons to study the causes and risk factors of breast cancer.

The causes of breast cancer in women are still unclear, and nonhereditary causes and risk factors predominate [7]. These include early menarche, hormone intake, nutrition, alcohol consumption, smoking, and obesity, all of which are commonly reported as risk factors [6]. Heredity accounts for only 5–10% of breast cancer cases, with germline mutations in BRCA1 or BRCA2 accounting for 30% of inheritable breast cancer cases [7]. Several researchers are working on the identification of specific genes as a cause of breast cancer, while others are conducting studies on the combinations of genes and other factors [8]. A study published in the Lancet in 2005 concluded that 21% of all breast cancer deaths worldwide were attributable to physical inactivity, overweight and obesity, and alcohol use [9]. An update from 2017 examined the connection between breast cancer incidence and environmental factors such as chemicals and radiation and found that exposure to these substances may lead to an increased risk of developing breast cancer [10]. However, numerous studies have identified late age of first full-term (if any) pregnancy, short periods of breast-feeding, dietary habits, quality and composition of meals, physiologic factors, lower age at menarche, and later menopause as causes and risk factors of breast cancer in women [11].

The risk and prevention panel from the 2012 Breast Cancer Campaign in the UK noted that the treatment of breast cancer have advanced, but efforts to predict which women are at an elevated risk and to prevent the disease have been less successful [12]. They estimated that in women at high and moderate risk, 50% of breast cancer could potentially be prevented by using current chemoprevention. Furthermore, they claimed that lifestyle measures, including moderating alcohol intake, exercise, and weight control, could reduce breast cancer risk in all women by approximately 30% [12]. Conversely, breastfeeding and physical activity are known protective factors [7]. Taken together, several factors are listed as causes of or risk factors for breast cancer (e.g., stress, hormone replacement therapy, vitamins and minerals, and active and passive smoking). Even though a large number of studies have identified genetic and many other risk factors for breast cancer, we do not fully understand what actually causes this disease. However, rather than writing about the causes, most studies focus on the identification of risk factors for breast cancer [11]. These include specific environmental factors that increase environmental toxicity and pollution and socioeconomic conditions such as rotating shift work and occupational exposure [11].

Reliable documentation addresses a social gradient in health. An individual’s health is progressively better the higher the socioeconomic position, and lower socioeconomic status increases an individual’s risk of chronic conditions [13]. Even so, in many studies, a higher breast cancer incidence has been found among women with higher levels of education and of higher socioeconomic status [14,15,16,17,18,19]. A systematic review and meta-analysis of socioeconomic inequalities in breast cancer in Europe found a significantly increased incidence and a significantly decreased case fatality for women of higher socioeconomic status [17]. The authors explained the decreased case fatality by possible differences in tumor characteristics, treatment factors, comorbidity, and lifestyle factors. It is therefore reasonable to assume that the way women live their lives may impact their risk of being diagnosed with breast cancer. Since women who are more highly educated tend to be healthier than women with lower levels of education [20], this phenomenon is of particular interest.

Among women affected by breast cancer, beliefs about the causes are not always consistent with the judgments of experts, such as breast density, reproductive history, alcohol consumption, physical inactivity, and age [21]. Instead, some breast cancer survivors believe that causes could include environmental factors, family history, fate, chance, or stress [21]. However, findings from a study designed to determine the levels of guilt and shame among patients with non-small-cell lung cancer compared to those with breast and prostate cancer show that a belief that one caused one’s own cancer is correlated with higher levels of guilt, shame, anxiety, and depression [22]. This evidence suggests that breast cancer patients experience shame and guilt.

In summary, breast cancer incidence rates are significantly higher among more highly educated women and women with higher socioeconomic status, and it is therefore reasonable to assume that the way women live their lives may impact their risk of being diagnosed with breast cancer. The causes and risk factors of breast cancer are not fully understood, yet there is a huge amount of research and, already, many existing systematic reviews on the topic. The aim of this overview of systematic reviews was to identify what is known about the causes and risk factors of breast cancer, exclusive of the genetic causes, and to investigate how the literature has identified and determined these risk factors.

## 2. Materials and Methods

### 2.1. Design

This is a systematic review and evidence synthesis which gives an overview of systematic reviews. Search on single studies on this topic showed many thousands of studies that had to be included; therefore, a systematic review of systematic reviews was necessary to retrieve a manageable number of studies for inclusion. The review is registered in PROSPERO (registration number: CRD42017062596) and was performed in accordance with the Preferred Reporting Items for Systematic Reviews and Meta-Analyses guidelines (PRISMA, 2020 checklist).

### 2.2. Search Strategy

In June 2016, a systematic literature search was conducted with the assistance of a research librarian. The selected databases were Embase, Medline, PsycINFO, and Cochrane. The following criteria were used to include relevant studies for this evidence synthesis: causes and risk factors of breast cancer, only systematic reviews or meta-analyses, only female participants, and lifestyle factors. The systematic reviews had to be published in the English language and in peer-reviewed journals over the previous 10 years. The exclusion criteria were male participants, conference abstracts, book chapters, editorials, comments, scientific statements, guidelines, protocols, publications on treatments, and genes as a risk factor or cause of breast cancer.

The search included Medical Subject Heading (MeSH) terms and keywords and variations of words related to causes of breast cancer and lifestyle factors such as psychosocial and sociocultural factors and socioeconomic status (see Figure 1). In addition, the term “protective factors” was used as an antonym because protective factors are often used to explain a possible connection to risk factors. The search was conducted in the following fields: title, abstract, heading word, table of contents, key concepts, original title, and tests and measures. We conducted three updated searches after June 2016, in January 2017, April 2019, and January 2020 (Figure 2).

### 2.3. Data Extraction

A data extraction form was created before the review to identify the key characteristics of the studies that met the criteria for inclusion. The following information was extracted from each included study: authors, publication year, country of included studies, aim, design and methodology of the study, time period of included studies, number of studies and participants included, results, conclusions, further research, limitations of the study, and the CASP score (Appendix A).

Four reviewers (BL, IHS, EKG, and IU) independently assessed all the studies and performed the quality scoring in four waves: (1) two reviewers independently screened the titles and abstracts of all articles and eliminated those not relevant for the purpose of the review; (2) if there were discrepancies among the findings of the two reviewers, they discussed with a third reviewer until a consensus was reached; (3) two reviewers independently screened the full-text articles and, based on the inclusion criteria, eliminated those not relevant; and (4) a third reviewer was involved if there were discrepancies among the findings of the two reviewers. The six Appendix A based on the information from the 122 included systematic reviews were made by a pair of authors, and the numbers of included studies and participants were counted by the reviewers and checked twice. If any discrepancy in the results was found by the two reviewers, then the two other reviewers checked the results until a consensus was reached.

### 2.4. Quality Appraisal of the Studies

Systematic reviews meeting the inclusion criteria were scored according to the Critical Appraisal Skills Program (CASP) (CASP, 2014) for systematic reviews, with some modification to match the needs of this study. The original checklist included 10 questions to help make sense of the systematic review. In the present review, we used the first seven questions. We asked: (a) if the review addressed a clearly focused question, (b) if the authors looked at the right type of papers, (c) if all the relevant important studies were included, (d) if the review’s authors did enough to assess the quality of the included studies, (e) if the results of the review had been combined, was it reasonable to do so, (f) what the overall results of the review were, and (g) how precise the results were. The last three questions (h, i, and j), addressing whether or not the results would help locally, were not included in the score.

### 2.5. Analyses

Part one of the analysis of all the included reviews consisted of a categorization of the articles in order to grasp an overview of the overall themes and different topics. We considered this process a qualitative content analysis [23], where six overall themes were extracted as areas possibly causing cancer. The second part of the analysis focused on the meta-analysis from all articles that reported effect size and whether they used a fixed-effect or random-effects model for the analysis and reported heterogeneity with I2. We constructed an SPSS file that included the reported effect size, confidence interval, heterogeneity, and whether the model used was fixed or random. Heterogeneity refers to the variation in outcomes between studies, and I2 can be interpreted as the percentage of the total variation which cannot be explained by coincidence. We divided the I2 score from the analyses into low (≤49.9% I2) or high (≥50% I2) heterogeneity [24,25]. Effect size is the magnitude of an effect and was reported by different measurements in the various studies (SRR, MD, SMD, OR, HR, RR, and ORR). We divided the results into high and low effect sizes, with high being over 10% and low being under 10% (≤9.9%). In addition, we separated analyses with effect size under 5% (≤4.9%), as one may argue that 10% is too high. The type of effect model used is important for the results of the analysis. When the heterogeneity is higher than expected, a random-effects model is preferred because it gives a wider confidence interval than a fixed-effect model [24,25]. When using random-effects weights as compared with fixed-effect weights, the extent of similarity will depend on the ratio of within-study error variance to between-studies variance [26]. Since we had to include as many as 122 systematic reviews with many different topics, an overview of the overall themes and different subcategories was needed (Table 1). Table 1 provides insight into the subcategories included in each category.

Further, all the significant findings in the analyses are illustrated in three diamonds (Figure 3). The first diamond shows the number of significant analyses divided into high and low effect size, high and low heterogeneity, and whether they used a fixed-effect or random-effects model. The second diamond shows the same analyses divided into overall themes, and the third diamond shows all the topics in the significant analyses.

## 3. Results

As shown in the flowchart (Figure 1), a systematic search with three updates identified 2387 systematic reviews. Of these, 406 articles were read in full text, and 122 systematic reviews were included in the present study, representing a total of 2404 single studies. The reviews included studies from all five continents (see Figure 4). Of the total number of single studies included, 77 were conducted by collaborators from two or more countries. The Nordic countries published 310 single studies, whereas Japan had 105 and China had 155 single studies. Several single studies were included in more than one systematic review.

The data enrollment ranged from 1942 to 2012, but only 36 systematic reviews specified the period when the enrollment of the data was carried out. The studies covered by the systematic reviews were published between 1954 and 2018 and included between 3 and 142 single studies in their meta-analyses. Only 12 of the systematic reviews did not perform meta-analyses. CASP scores ranged from four out of seven (*n* = 1) to seven of seven (*n* = 58). Of a total of 122 systematic reviews, 53 were conducted in China.

All included systematic reviews were categorized into six different topics: (1) influences from external factors, 17 (13.9%) [27,28,29,30,31,32,33,34,35,36,37,38,39,40,41,42,43], (2) fertility and drugs, 26 (21.3%) [44,45,46,47,48,49,50,51,52,53,54,55,56,57,58,59,60,61,62,63,64,65,66,67,68,69], (3) alcohol and tobacco, 10 (8.2%) [70,71,72,73,74,75,76,77,78,79], (4) lifestyle, physical activity, and body size, 17 (13.9%) [80,81,82,83,84,85,86,87,88,89,90,91,92,93,94,95,96], (5) nutrition, 31 (25.4%) [97,98,99,100,101,102,103,104,105,106,107,108,109,110,111,112,113,114,115,116,117,118,119,120,121,122,123,124,125,126,127], and (6) blood and metabolism, 21 (17.2%) [128,129,130,131,132,133,134,135,136,137,138,139,140,141,142,143,144,145,146,147,148] (For more details see Appendix A). As shown in Figure 5, three of the studies about alcohol and tobacco were published in 2007, and no studies are included in this topic from 2018 or 2019. The number of studies on blood and metabolism increased from 2014, and, in this category, only two studies were included from before 2014. The same tendency was found in the topic of nutrition, where the number of studies increased rapidly from 2014, with 25 systematic reviews and meta-analyses published after 2014. Most of the reviews on lifestyle, physical activity, and body size were published from 2014 to 2017, and studies on external factors increased from 2012 to 2014 but did not appear much in the other years. The category related to fertility and drugs included some published studies from 2008 to 2010 and peaked in 2015 and 2018.

Table 1 shows the different fields in each topic; for the most part, there was only one systematic review covering each theme. However, six systematic reviews about red and processed meat were included, five about body mass index, and four each about cadmium, physical activity, adiponectin, vitamin D, and working night shifts. A total of 236 analyses were gathered from the 122 systematic reviews. The analyses conducted in each review varied from 1 to 38. There were 123 analyses with significant findings, but 14 of these analyses did not show the percentage of heterogeneity in I2. Figure 3 (“The Diamonds”) shows the findings of 109 meta-analyses related to risk increase and protective factors for breast cancer. Table 2 shows the 36 analyses with the highest effect size and low heterogeneity. High breast density has the highest effect size for increasing the risk of breast cancer, and a high sex-hormone-binding globulin (SHBG) level is the most protective factor in decreasing breast cancer risk according to our criteria. There were only two studies of food and dietary patterns in the “top list”; meat consumption more than three times a week and fat intake increased the risk of breast cancer. Ten of these 36 meta-analyses were from only one systematic review [95]. Seven of these were about different types of physical activity, which were found to be inversely associated with breast cancer risk. The protective association varied from 0.79 (I2 = 6%) to 0.90 (I2 = 24%). As shown in Figure 3, analyses with different types of physical activity were found in all places in the diamond, except in studies with low effect size and low heterogeneity (I2) made by a fixed-effect model.

## 4. Discussion

In this systematic review and evidence synthesis, the results were confusing and conflicting. We investigated what is known about the causes and risk factors of breast cancer, exclusive of genetic causes. Since findings from previous studies have shown that breast cancer in women does not follow the socioeconomic gradient in health [14,15,16,17,18,19], it was of interest to explore whether lifestyle factors or the way women live their lives may explain this inverse tendency. In this study, 224 meta-analyses from 122 systematic reviews were categorized into six main areas. More than 30 subcategories may be considered triggers, risk factors, or causes of breast cancer for women. Rather than finding the causes of breast cancer in the literature, we mostly reviewed risk factors. Although we initially searched for “causes”, we ended up with 95 titles of systematic reviews and meta-analyses with risk factors. None of the titles that met the inclusion criteria included the term “causes”. We regard this as a very important finding. To investigate what is known in the research literature about the causes of breast cancer is demanding and challenging work. According to the criteria defined in the method, we created a “top list” with the 36 most important findings from the meta-analyses (Table 2). Increased breast density and bone density, never having married, higher BMI, and meat consumption were risk factors for breast cancer, and high sex-hormone-binding globulin levels, progestin, physical activity, and higher BMI were the most protective factors, and at the top of our list.

Except for never having married, these findings are in line with what is already known as causes and risk factors for breast cancer [6,9,11]. It is interesting that being unmarried is in second place in our “top list”. It is not usually described as a risk factor or cause of breast cancer, but the lifestyle associated with being single may be a risk factor for some women. A previous study found that age at first full-term pregnancy accounted for most of the association between higher education and invasive breast cancer [18]. On the other hand, it is surprising that alcohol consumption was not in the “top list” of risk or causes of breast cancer, since alcohol consumption is listed as a risk factor [6], and deaths from breast cancer are linked to alcohol use [9]. Three systematic reviews of alcohol use and breast cancer are included in this evidence synthesis, two with meta-analyses, and it is interesting to note that the findings are inconsistent. The conclusions varied from “there is an association” between alcohol consumption and breast cancer risk [78], to “the association remains insufficient” [77], to “high intake of wine contributes” to breast cancer risk, but “protection is exerted with low doses of wine” [76]. Although one of these meta-analyses concluded with an association with an effect size of 1.28, the result did not qualify for our “top list” because of high heterogeneity (I2 = 73.5%) [78]. In addition, in the systematic review without a meta-analysis, researchers found a positive association in one study, while another study showed a nonsignificant inverse association [77]. The authors of this systematic review reported low methodological quality and a low number of studies included.

However, low quality may also be an issue in our systematic review since only 57% of the included systematic reviews used a quality assessment to evaluate the included studies. It is hard to draw conclusions without having this quality information. Nevertheless, it might imply that it is difficult to determine whether or not these studies have deficiencies in methodology. Although the purpose of systematic reviews is to give an overview and balanced picture of what research has shown about a specific issue, it is often difficult to draw conclusions from all results in the same direction.

Indeed, in the present evidence synthesis, there were also conflicting and contradictory findings in several of the meta-analyses. As seen in Table 2, findings from one study showed that a high BMI increased the risk of breast cancer [93], and findings from another study showed the exact opposite, that a higher BMI reduced the risk for breast cancer [92]. The authors of the study do not provide any explanation for this contrary result [92]. Both studies looked for risk factors increasing breast cancer in women. The first study investigated risk factors in the Eastern Mediterranean region, and the second only included women 40–49 years of age. Of the included studies 81% performed subgroup analyses and looked for risk factors in, for example, age categories or in different parts of the world.

Six systematic reviews about red and processed meat and increased risk of breast cancer were included in the present study. Findings varied from “there is an association” in premenopausal women [104], to a “weak” positive significant association for each 100 g increase of red meat in postmenopausal women, to “red and processed meat intake does not appear to be independently associated with increased risk of breast cancer” [103]. A study in 2018 confirmed that red and processed meat intake increased the risk of breast cancer [107], while another study concluded that a “high consumption of processed meat was associated with higher overall postmenopausal breast cancer”, and that red meat consumption was not associated with breast cancer [105], while a third study from the same year found “an association” between red meat consumption and increased breast cancer risk in premenopausal women [106].

In summary, the results of these studies are inconsistent, and their interpretations are difficult to understand. The authors “indicate” and “suggest” in the description of results and conclusions, although the heterogeneity (I2) was high, and the analysis was weak and nonsignificant in some of the studies. The results in these latter studies are not convincing due to methodological considerations, although many of the same studies are included in several systematic reviews and meta-analyses. Overlapping meta-analyses often can be confusing because they may reach different conclusions, and the interpretation of even the same results can differ across systematic reviews and meta-analyses on the same topic [149]. A previous study showed that of 73 eligible meta-analyses published in 2010, 49 (67%) had at least one other overlapping meta-analysis [150]. Most of the included analyses in the present review had an I2 > 50%, and in that case, a fixed-effect model is not preferred. However, it is a bit surprising to note that six of 20 meta-analyses using a fixed model appear on the “top list” (Table 2). Out of these six analyses, three were from only one of the 122 systematic reviews.

Some researchers explain the challenges related to confusing results and interpretations with the lack of guidelines and recommend checking if the included studies used the same questions, if they had the same quality and selection criteria, etc., or if their interpretations of similar results were weak [151]. Our findings from the studies on red and processed meat confirm some of these challenges. A previous study stated that the interpretation of even the same results can differ across systematic reviews and meta-analyses on the same topic, especially when the authors have strong motivation to reach specific conclusions [149]. This last statement is difficult to examine in detail and was not explored in our study.

Given its authority in the field, it is not surprising that in our “top list” there are ten analyses from an update from World Cancer Research Fund International [95]. These analyses indicate associations between physical activity in adulthood, BMI, waist circumference, and risk of breast cancer. Five of these meta-analyses show that different types of physical activities protect women against an increased risk of breast cancer [95]. Two other analyses on the “top list” confirm these results [80,81]. However, as shown in Section 3, there are also several studies about exercise and physical activity with high heterogeneity (I2), which did not satisfy the requirements to be on our “top list” (Figure 3).

Although nutrition was the area with the most systematic reviews and meta-analyses (*n* = 32), only meat consumption and fat intake together with iron intake are found in our “top list” to increase the risk of breast cancer in women. The fact that the number of systematic reviews and meta-analyses in the area of nutrition has increased sharply over recent years, without important findings toward causes or risk factors, may raise the question as to whether nutrition is the right area to explore in order to identify causes and risk factors of breast cancer in women. However, how do we choose which paths of inquiry to take in order to find out why breast cancer seems to be more likely among more highly educated women and those at a higher socioeconomic level? It will largely be based on what we already know and what we hypothesize in combination with what characterizes highly educated women and women with higher socioeconomic status.

Since it is difficult to determine the causes and risk of breast cancer, one should caution against stating that lifestyle factors such as high BMI, smoking during pregnancy, consumption of fatty food, or intake of red meat are the causes. As this review states, we do not actually have precise knowledge of the causes of breast cancer. However, health professionals who think they know (and thereby conclude from assumptions) can inflict more pain on people who are already distressed. This issue is important because many women who have developed breast cancer blame themselves for the situation and incorrectly assume that the breast cancer has been triggered by their smoking, diet, body weight, use of alcohol, lack of physical exercise, or any other factor in their life. In addition, when the risk and causes of breast cancer are associated with lifestyle choices, posted on social media, and taken for granted in descriptions of public health, these women may be stigmatized [152].

When it comes to the question of higher education and higher socioeconomic status as triggers of breast cancer, it is often observed that women with higher education are overrepresented among breast cancer victims [14,15,16,17,18,19]. However, there is no clear and convincing evidence in our study to indicate any causal relationship between breast cancer and higher education or higher socioeconomic status and breast cancer. As Oscar, 15 years old, said; “It’s really strange that Mom got breast cancer, she exercised quite a lot and ate healthy” [153].

### Strengths and Limitations

The strengths of this study are the thorough and transparent processes of searching, evaluating, and synthesizing the systematic reviews included in this evidence synthesis. Performing such a review of systematic reviews leaves room for interpretation, which might be a bias. Additionally, a limitation might be linked to the search process and the selection of the included systematic review studies. However, support from a trained research librarian should compensate for this possible limitation regarding the search process by secure a high-quality literature search. Another limitation is that we did not include genes in the search but included studies that included genetic factors such as sex-hormone-binding globulin.

## 5. Conclusions

Six overall themes and more than 27 topics were identified. In our “top list” of the 36 most important findings, high breast density had the highest effect size for increasing the risk of breast cancer, and a high sex-hormone-binding globulin level was identified as the most protective factor in decreasing breast cancer risk. Notwithstanding the comprehensive work in performing these studies, there is still a large degree of uncertainty with respect to what exactly triggers or may influence or cause breast cancer. Unfortunately, this extensive work has not provided a clear and convincing answer as to why women develop breast cancer. Based on our review, however, it is important to point out this lack of consensus because the recommendations from some health authorities use several uncertain consensus statements to guide us to a healthy lifestyle. We therefore argue that this review of systematic reviews provides a valuable contribution to the research portfolio regarding breast cancer in women. We also argue that it highlights that health professionals should be careful because there still remains a great deal of uncertainty regarding the risk factors and causes of breast cancer in women.

## Figures and Tables

**Figure 1 cancers-16-01583-f001:**
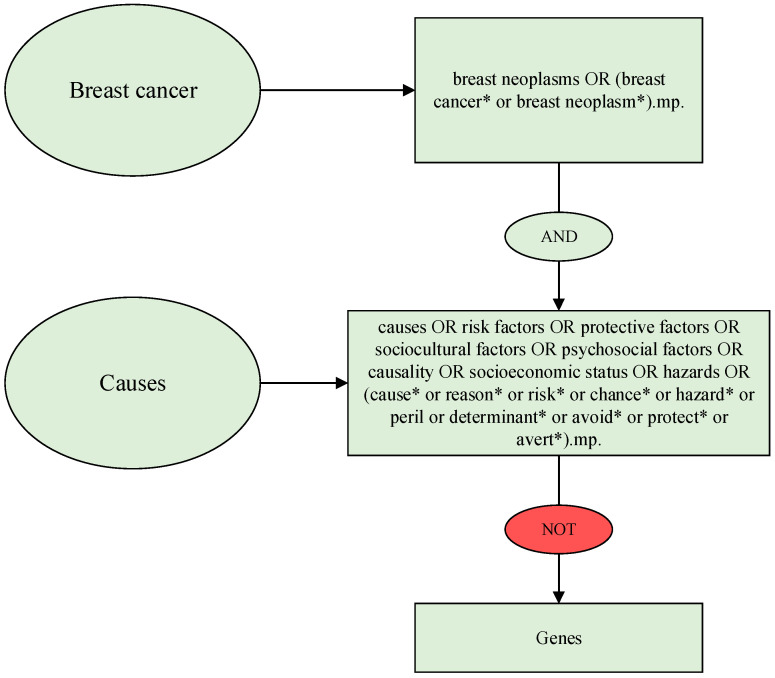
Extracts of MeSH terms, keywords, and words used in the literature search. * symbol for truncation.

**Figure 2 cancers-16-01583-f002:**
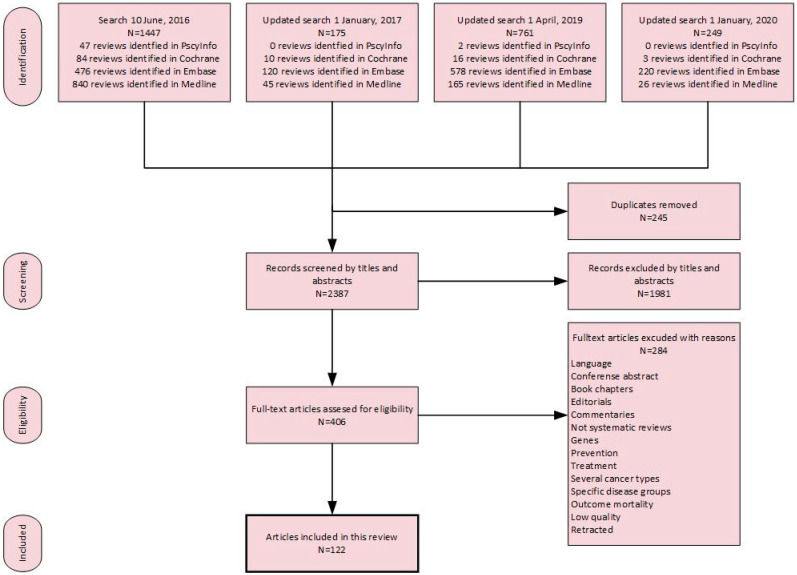
Flowchart of number of results from the literature search.

**Figure 3 cancers-16-01583-f003:**
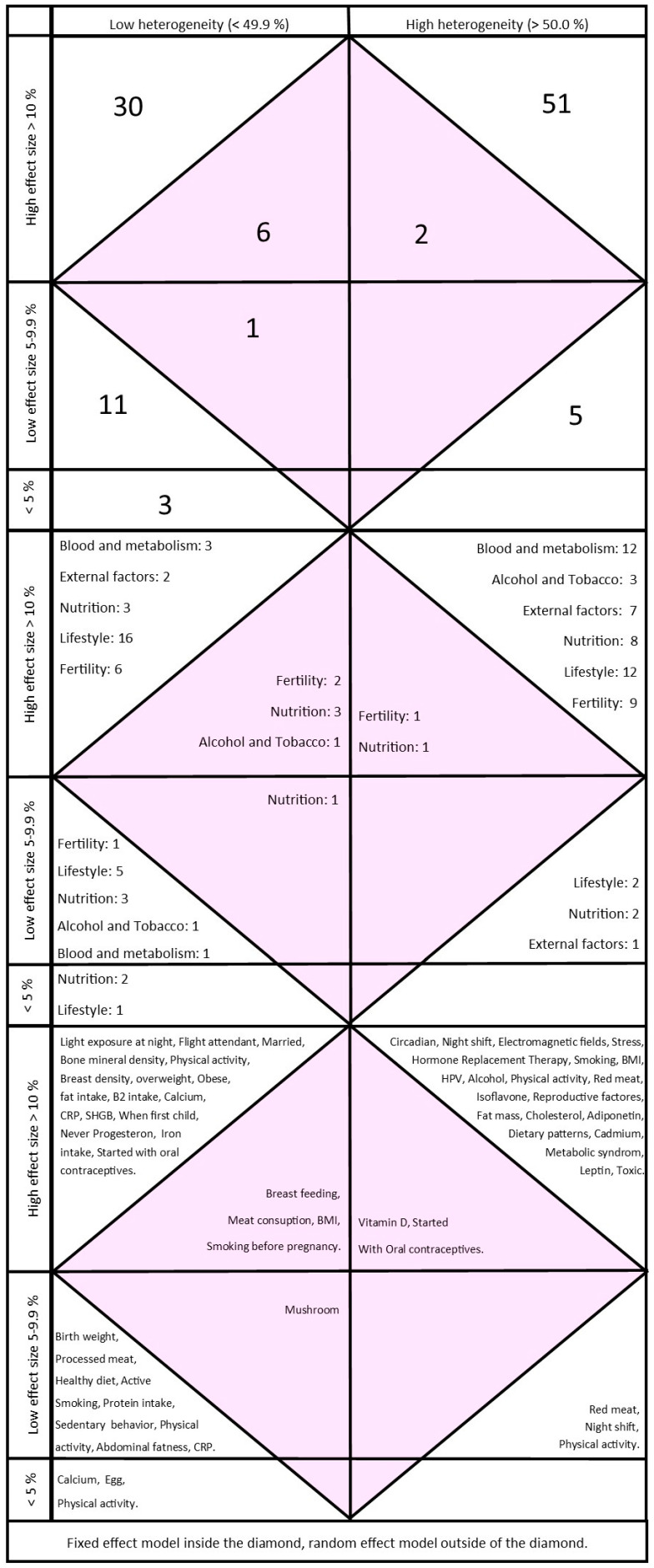
“The Diamonds”. Analyses of meta-analyses divided into high and low effect size, high and low heterogeneity, and whether they used a fixed-effect or random-effects model: The top diamond shows the numbers of studies, the middle diamond shows the themes, and the bottom diamond shows the topics.

**Figure 4 cancers-16-01583-f004:**
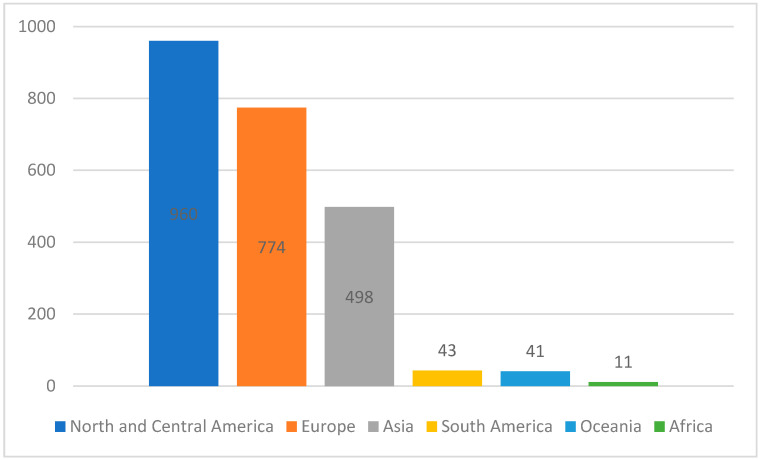
Bar graph: Illustration of where the primary studies included in the 122 systematic reviews had been conducted.

**Figure 5 cancers-16-01583-f005:**
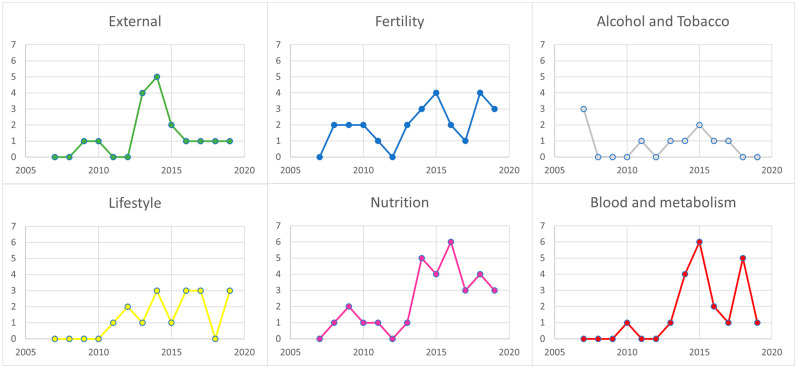
Year of publication: Graphs illustrating which year each included systematic review was published, divided into the six main themes.

**Table 1 cancers-16-01583-t001:** Category, subcategories, and numbers, of the included articles.

Category:	Subcategories:	Number:	Total:
Alcohol and tobacco	Passive and active smoking	3	10
	Passive smoking	3	
	Smoking before pregnancy	1	
	Alcohol	1	
	Wine	1	
	Long-term alcohol intake	1	
Lifestyle, physical activity, and body size	Physical activity	4	17
	Sedentary behavior	1	
	BMI	5	
	Obesity and overweight	2	
	Abdominal fatness	1	
	Breast size	1	
	Breast density	1	
	Several lifestyle factors	2	
Influences from external factors	Night shift	4	17
	Circadian	1	
	Light expose at night	1	
	Sleep duration	2	
	Flight attendant	1	
	Electromagnetic fields	2	
	Polychlorinated biphenyl	1	
	DDE	1	
	Stress	3	
	Striking life events	1	
Blood and metabolism	Adiponectin	4	21
	Leptin	1	
	Vitamin D	4	
	Calcium	1	
	Vitamin A	1	
	Metabolic syndrome	2	
	CRP	2	
	Cholesterol	1	
	Lipid	1	
	Blood type	1	
	Oxidative stress	1	
	Toxic elements	1	
	Sex-hormone-binding globulin	1	
Fertility and drugs	HPV	1	26
	Intrauterine environment	1	
	Abortion	1	
	Breast feeding	3	
	Preeclampsia	2	
	Birth Weight	1	
	In vitro fertilization	3	
	Hormone replacement therapy	2	
	Progesterone	3	
	Estradiol	1	
	Oral contraceptives	2	
	Folate	2	
	NSAIDs	1	
	Antibiotics	1	
	Bone mineral density	2	
Nutrition	Red and processed meat	6	31
	Mushrooms	1	
	Black Cohosh	1	
	Black tea	1	
	Eggs	1	
	Coffee	1	
	Multivitamins	1	
	Calcium	1	
	Iron	1	
	Isoflavone	1	
	Overall diet	1	
	Dietary patterns	3	
	Cholesterol	1	
	Total and serum fat	2	
	B2	1	
	Protein	1	
	Inflammatory potential	1	
	Glycemic index and load	2	
	Cadmium	4	
Total:		122	122

**Table 2 cancers-16-01583-t002:** “Top list”. The most important findings, 36 meta-analyses with high effect size and low heterogeneity (I2).

Increased Risk			Reduced Risk		
**First Author**	**Effect**	**Theme**	**First Author**	**Effect**	**Theme**
Bae [91]	3.23	Breast density	He [147]	0.64	SHGB
Vishwakarma [68]	2.29	Never married	Asi [58]	0.67 †	Progestin
Namiranian [93]	2.21 *	BMI > 30	Unar-Munguia [50]	0.72 *	Breast feeding
Qu [66]	1.82	Bone mineral density	Nelson [92]	0.74	BMI > 30 woman 40–49 years
Namiranian [93]	1.71 *	BMI 25–30	Chen J.H. [67]	0.75	Bone mineral density
Qu [66]	1.62	Bone mineral density	Chan D.S.M. [95]	0.79	Vigorous activity premenopausal
Bae [91]	1.62	Breast density	Wulaningsih [137]	0.80 †	Calcium
Vishwakarma [68]	1.57	Age when having first child	Chan D.S.M. [95]	0.85	Adult weight loss premenopausal
Liu [43]	1.40	Flight attendant	Yu [116]	0.85	B2 intake
Namiranian [93]	1.39 *	Meat consumption more the three times a week	Chan D.S.M. [95]	0.86	Early adult BMI per 5 kg/m
Yang W.S. [33]	1.17	Light exposure at night	Chan D.S.M. [95]	0.86	Total physical activity postmenopausal
Chan D.S.M. [95]	1.17	Gain in BMI per 5 kg/m^2^. postmenopausal	Nelson [92]	0.86	BMI 25–30-woman 40–49 years
Guo [140]	1.16	C-reactive protein	Chen X. [81]	0.87	Physical activity
Ji [69]	1.16 *	When started with oral contraceptives	Wu Y. [80]	0.88	Different physical activity
Chang V.C. [127]	1.12	Iron intake	Chan D.S.M. [95]	0.88	Recreational activity postmenopausal
Chan D.S.M. [95]	1.11	Waist circumference pr 10 cm postmenopausal	Chan D.S.M. [95]	0.90	Adult weight loss postmenopausal
DeRoo [75]	1.10 *	Smoking before pregnancy	Chan D.S.M. [95]	0.90	Vigorous activity postmenopausal
Cao [114]	1.10	Fat intake	Chan D.S.M. [95]	0.90	Occupational activity postmenopausal

* Fixed-effect model. † No subgroup analysis.

## Data Availability

No new data were created or analyzed in this study. Data sharing is not applicable to this article.

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
