# Peer review of "Causes and Risk Factors of Breast Cancer, What Do We Know for Sure? An Evidence Synthesis of Systematic Reviews and Meta-Analyses"

_cancers, 2024, doi:10.3390/cancers16081583_

Round 1
Reviewer 1 Report
Comments and Suggestions for Authors
It was a very interesting review.
A few questions to the authors:
1. Menopause is not identified as a separate risk factor for breast cancer in any of the studies?
2. In developed countries, the incidence of breast cancer is higher, why is there no data in the review on education and type of work as risk factors for breast cancer?
3. In Figure 3, it is better to place the diamonds vertically, since the inscriptions on the figures are unreadable.
4. In the table with risks, many factors are repeated by different researchers, if they are grouped, then the prominent risk factors are clearly visible - obesity, for example (BMI more than 30, 25-30...), some factors require clarification - were not married - this is not Were you pregnant and didn’t have children or is it more about lifestyle? Well, the opposite factors in two columns attract attention: bone density and calcium - in fact, these are the same risk factor. It seems to me that we also need a generalizing table with average risks for aggregated categories based on Table 2.
Reviewer 2 Report
Comments and Suggestions for Authors
The weakest part of this manuscript is that the data and conclusion will not benefit future breast cancer research and patients.
Comments:
1. Table 1 needs re-constructure. Not easy to read and understand the contents and significance.
2. On Table 1: authors need to add breast/nipple appearance as one of the risk factors.
3. On Figure 4: please explain why Australia and New Zealand have few studies used in this manuscript.
4. On Figure 4: Will bar graph be better?
5. On Figure 4: Any geologic difference on the risk factors of breast cancers?
6. On Figure 5: what is external?
7. On Figure 5: Is it important overall to show these 6 main themes vs years?
Round 2
Reviewer 1 Report
Comments and Suggestions for Authors
I have no further comments on the manuscript.
Reviewer 2 Report
Comments and Suggestions for Authors
The manuscript improves significantly. No more comments.